# Compressive Strength Properties Perpendicular to the Grain of Hollow Glue-Laminated Timber Elements

**DOI:** 10.3390/polym14163403

**Published:** 2022-08-19

**Authors:** Nikola Perković, Jure Barbalić, Vlatka Rajčić, Ivan Duvnjak

**Affiliations:** Structural Department, Faculty of Civil Engineering, University of Zagreb, 10000 Zagreb, Croatia

**Keywords:** timber, compression strength, perpendicular to grain, glulam, innovative, hollow, FEM

## Abstract

Timber is one of the fundamental materials of human civilization, it is very useful and ecologically acceptable in its natural environment, and it fits very well with modern trends in green construction. The paper presents innovative hollow glued laminated (GL) timber elements intended for log-house construction. Due to the lack of data on the behavior of the hollow timber section in compression perpendicular to the grain, the paper presented involves testing the compression strength of elliptical hollow cross-section glue-laminated timber specimens made of softwood and hardwood, as well as full cross-section glue-laminated softwood timber specimens. The experimental research was carried out on a total of 120 specimens. With the maximal reduction of 26% compared to the full cross-section, regardless of the type of wood and direction of load, the compression strength perpendicular to the grain of hollow specimens decreases by about 55% compared to the full cross-section, with the coefficient k_c,90_ equal to 1.0. For load actions at the edge and the middle of the element, k_c,90_ factors were obtained with a value closer to those obtained for full cross-section, which indicates the same phenomenology, regardless of cross-sectional weakening. At the same time, the factors in the stronger axis are lower by about 10%, and in the weaker axis by about 30% compared to those prescribed by the Eurocode. Experimental research was confirmed by FEM analysis. Comparative finite element analysis was performed in order to provide recommendations for future research and, consequently, to determine the optimal cross-section form of the hollow GL timber element. By removing the holes in the central part of the cross-section, the stress is reduced. The distance of the holes from the edges defines the local cracking. Finally, if the holes are present only in the central part of the element, the behavior of the element is more favorable.

## 1. Introduction

Timber is a renewable, biodegradable, and environmentally friendly material that absorbs carbon dioxide from the atmosphere. During manufacturing, it requires little energy, it opens all kinds of new possibilities during operation in wooden structures, where the hitherto widespread use of concrete and bricks can be replaced. Based on all the above, it can be concluded that the construction industry is increasingly turning to timber as a construction material. Accordingly, there are numerous innovations and the production of factory wooden elements such as multi-layer cross-laminated timber (CLT—cross laminated timber), laminated veneer lumber (LVL—laminated veneer lumber), cross-glued veneer board (plywood), parallel glued " veneer noodles (PSL—parallel strand lumber), parallel glued wood "noodles" (LSL—laminated strand lumber), boards with oriented chipboard (OSB—oriented strand board), parallelly oriented chipboard (PLS—parallel strand board), chipboard boards (particleboard), fiber boards (—HB, MBH, MDF).

A typical log-house (or log-haus, Blockhaus, etc.,) system represents a traditional construction system widely used in northern regions as well as in urban regions with a high seismic hazard such as the Mediterranean area [1]. The basic timber wall components vertically stacked one upon another are recognized as very efficient and reliable timber structures. The constructive principle of log-house is represented by the superposition of a series of timber logs, although several adapted construction systems were developed recently by Perkovic et al. [2].

In order to achieve an ecological approach, the idea of assembling a hollow glue-laminated timber wall element from slats, which were the waste product in process of carpentry production, was developed by Croatian company TERSA Ltd. Slats are arranged 20 mm thick and 120 mm in width lamellas. The lamellas are profiled before gluing into a single laminated beam, in such a way that combined creates an ellipse-shaped perforation between lamellas in the cross-section of 120 mm in width and 240 mm in height. This type of cross-section guarantees better behavior during the moistening and drying of wood, as well as better energy characteristics of load-bearing elements.

Although these elements have reliable properties [2], many structural aspects need further recognition. In the design of log-house structures with hollowed elements, one of the most important problems is the proper verification of the stress state at the places of concentrated force inputs, but also the bottom of the wall. Therefore, there is a need to evaluate the compression stresses and deformations perpendicular to the grain for different boundary conditions and cross-section axes orientations of hollow elements. Selected results from testing wall members are presented. The test results were compared with the current approach from Eurocode 5 [3].

### 1.1. State-of-the-Art

One point of discussion in the scientific community was whether standards should aim to maintain either well-defined basic material properties or reflect typical uses. Europe has opted for the former (the scientific) approach, on the assumption that it would then be possible to calculate the behavior in practical application situations, while US/Canada and Australia/New Zealand have chosen the latter (the technological) [4].

The compression perpendicular to the grain design approach presented in Eurocode 5 [3] is based on experiments by Madsen et al. [5]. Some modifications, as currently valid, were proposed by Blass and Görlacher [6]. According to this model, the load-bearing capacity of the element is obtained from effective contact area A_ef_, characteristic compressive strength perpendicular to the grain f_c,90, k,_ and the factor k_c,90_, which considers the load configuration, the possibility of splitting, and the degree of compressive deformation [3]. The effective contact area A_ef_ should be determined considering an effective contact length parallel to the grain, where the actual contact length, l, at each side is increased by 30 mm.

According to EC5 [3], the value of k_c,90_ should be taken as 1.0, unless the conditions in the following paragraphs apply. In these cases, the higher value of k_c,90_ specified may be taken, with a limiting value of 1.75. For members on continuous supports, provided that *l_1_* ≥ 2 *h* (see Figure 1a), the value of k_c,90_ should be taken as 1.25 for solid softwood timber and 1.5 for glued laminated softwood timber, where *h* is the depth of the member and *l* is the contact length. For members on discrete supports, provided that *l_1_* ≥ 2 *h* (see Figure 1b) the value of k_c,90_ should be assumed to be 1.5 for solid softwood timber and 1.75 for glued laminated softwood timber if *l* ≤ 400 mm. Leijten et al. [4] pointed out the inconsistencies of the mentioned discontinuities and determined the coefficient k_c,90_ based on empirical results. Finally, they proposed modified expressions for k_c,90,_ using the physical model of Van der Put [7].

The compressive strength of wood in the direction perpendicular to the grain, *f_c,90, k_*, (CSPG) plays an important role and frequently governs the structural design. Obviously, CSPG depends on the type of wood and varies in radial and tangential directions [8,9,10]. Hoffmeyer et al. [11] concluded that the combined role of tensile and shear stresses perpendicular to the grain occurs in compression specimens of solid as well as glued laminated wood, where, at the design level, the 5% characteristic strength is not significantly different. Gehri [12] presented a study to verify the relationship between compressive strength and wood density, which is particularly evident when comparing healthy wood to rotten or insect-deteriorated wood [13]. Although wood is recognized as a building material due to highly technical-material characteristics in the direction parallel to the grains, it is necessary to note that elongated cells of the wood are stiffer and stronger when loaded along the axis of the cell rather than when loaded across it [14,15]. So, the modulus of elasticity in the direction perpendicular to the grain decreases by 30 times and strength by eight times for softwood and three times for hardwood [16].

To determine the real f_c,90, k_ value, the European (CEN) testing standard EN408 prescribes a method in which a block of timber is loaded in uniform compression over its entire surface. On the other hand, the American (ASTM) test standard D143 is based on the approach in which the test piece is a timber block, and the load is applied in the middle through a steel plate, where the test is primarily intended to simulate the behavior of a wood joint resting on a wall or foundation and does not intend to determine a physically correct perpendicular to grain strength [4]. In the absence of any physical model to modify the results and to account for situations deviating from the test set-up, modification factors were established and reported by Kunesh [17]. Madsen et al. [18] also took an interest in the relationship between deformation and compressive strength and recognized shortcomings of the ASTM method. Furthermore, Leijten [19] pointed out that in the Scandinavian countries, the standard characteristic bearing strength for a spruce wood element has double or even triple value than the stress at the proportional limit determined by tests, making values reported in the European standards questionable and very conservative. Further investigation is presented in [20]. Considering the above, the problem of a unified approach to determining the standard strength is obvious.

As a special issue, it should be highlighted the compression strength perpendicular to the grain in cross-laminated timber (CLT), where significant conclusions are given in [21,22,23,24,25,26].

A standard European test procedure for the determination of CSPG is defined by standard EN408 [27]. This procedure is based on former prescriptions of the fiber stress at the proportional limit, or the stress which causes a 1% deformation, first presented by Kolmann and Côté [28]. Using the test results, on the plot load/deformation (F-∆h) curve, a line (1) parallel to the linear part of the load-displacement and determined by values of 0.1 F_c,90, max,_ and 0.4 F_c,90, max_ as intersections with the curve, needs to be defined. Finally, the ultimate load capacity, F_c,90, max_ is defined as the intersection of curve and line (2), which is offset by 1% of the standardized specimen depth h and parallel to the line (1). The force corresponding to the upper limit of the linear segment of the load/displacement (F-∆h) curve is known as the proportional limit F_c,90, prop_ [29].

### 1.2. Objectives

The main idea for this research came from the doubt about the sufficient bearing capacity of hollow elements to the compression perpendicular to the grain. Although the arrangement of the cavities is designed to ensure a regular force path to the support, the strength properties necessary for design could not be determined just on the wood class. In Figure 2, one can see the constitutive elements-lamella, and finally, the assembled cross-section of the innovative hollow glued laminated timber element. 

The main objective of this research was to investigate the load-carrying capacity in the compression perpendicular to the grain of hollow glue-laminated softwood and hardwood timber element and to compare it with the requirements of the current European standard for full cross-section elements. According to the possible orientation of the elements, and thus of the cross-section orientation, it was necessary to test the specimens in both, strong-axis, and weak-axis directions. In order to compare the compressive strength and factor *k_c,90_*, taking into account the load configuration, the possibility of splitting, and the degree of compressive deformation, depending on support and load type, several other test set-ups involving loading of the specimens’ proportional rectangular surface only at the edge parts, as well as only at central parts, was investigated. 

Variant combinations of lamellas allow for different percentages of perforation of the cross-section. In this paper, only the maximally perforated variant with elliptical holes was investigated and compared with normal (full) timber elements. Compared to the full GL elements, the cross-section area was reduced by 26%. Other variants were investigated by finite element modeling, with the goal of finding the optimal layout of the holes regarding stress distribution.

The elements are normally produced in two versions, made of softwood with the predominant use of European fir (*Abies alba*), and hardwood with the predominant use of European hornbeam (*Carpinus betulus*). Both types of hollow elements were tested, as well as full cross-section elements made of softwood, in a total of 120 specimens.

There is evidence that metal-to-wood compression inaccurately reflects typical wood-to-wood compression often present in structural applications [29,30]. However, a common method using metal-on-wood compression [10,31,32] was not applied. Instead, between the metal and the specimen, hardwood elements with prescribed contact surfaces were inserted. Although the digital image correlation (DIC) technique in the testing of structural elements has already been proved [33,34,35] for a better insight into the redistribution of stress, in this case, it would be useful only for certain types of samples. Therefore, it was not used in this test.

## 2. Materials and Methods

### 2.1. Test Setup

Preliminary research was done by Perkovic et al. [2], where different types of cavities were investigated, as well as the layout of the cavities themselves. In addition to elliptical cavities, the behavior of samples with circular holes was also investigated, and the conclusion was that the samples with circular cavities had significantly lower load-carrying capacity and less favorable failure modes. Consequently, the continuation of the research was carried out only with elliptical holes, as well as a modified cavities layout. First, this refers to the first and last lamella, which is shaped differently from the inner lamellas, with the aim of increasing resistance, considering that the highest normal stresses occur at the edges of the cross-section. Furthermore, it was concluded that the more favorable arrangement of the holes is such that they are set in columns, that is, that there is a “web” over which the load can be transferred from the top to the bottom. In addition, the type of adhesive was changed, considering that in the previous investigation [2], the fracture mode occurred in many samples due to the adhesive line, and thus the consistency of the results was disturbed. In this research, a PUR adhesive, *Kleberit 510*, intended for load-bearing timber structures, was used [36].

European standard EN 408 [27] was used for the CSPG evaluation. The production of the test specimens was designed so they match the actual shape of the element, and at the same time meet all the conditions prescribed by the standard. The loaded surfaces were carefully prepared to ensure that they are flat and parallel to each other and perpendicular to the axis of the test specimens. This preparation was performed after conditioning the timber. In the case of glued laminated elements, the test specimens provided for determining the base value of CSPG, are assigned in accordance with EN 408 [27]. In the case of glued laminated elements, height h of 200 mm, minimum width b_min_ of 100 mm, and the surface that is fully loaded b × l of 25,000 mm^2^ is defined, to achieve a volume of 0.01 m^3^ for the tested specimens. In addition to the specimens prescribed by the standard, additional specimens were defined and loaded on the edge and in the middle part of the element, in order to determine the distribution of force along the specimens.

The specimens were mounted vertically between the steel plates of the testing machine and the appropriate compression load. Due to the indentation of the end lamellas when the load is acting in a strong-axis direction, additional timber elements were made for this purpose, which on the one end corresponding to the indentation on the sample, and on the other end are flat and thus enable the introduction of loads over the entire surface. Here, the stronger axis represents the axis along which the lamellae are arranged. The length of the gauge, h_0_ (approximately 0.6 h), is located centrally in the specimen height and no closer than b/3 of the loaded ends of the specimen, as shown in Figure 3.

The loading equipment used can measure the load to an accuracy of 1% of the load applied to the test specimen or, for loads less than 10% of the maximum load, to an accuracy of 0.1% of the maximum load. The universal testing machine Z600E with a capacity of 600 kN was used for testing. The test specimen has been loaded without eccentricity, which was achieved using spherically seated load heads. According to the standard [27], displacement control was used at different speeds from 3 to 6 mm/min, depending on the material and the position of the sample (loaded in strong-axis or weak-axis direction). The loading rate has been adjusted so that the maximum load F_c,90, max, est_ or F_c,90, max_ was reached within (300 ± 120) s. The test was stopped after reaching the compressive strength of the timber elements. This rate was determined from the results of preliminary tests.

The compressive strength f_c,90_ is determined from the equation:f_c,90_ = F_c,90,max_/bl,(1)

f_c,90_—compression strength (N/mm^2^)F_c,90,max_—maximal compression load parallel to the grain (N)b—width (mm)l—length (mm)

Compressive strength is calculated to an accuracy of 1%.

### 2.2. Type and Number of Samples

To fully consider the behavior of innovative hollow glue-laminated timber elements, all combinations of specimen positions and loads were investigated. This refers to the positioning and loading of specimens in both the strong- and weak-axis directions, and the position of the force with respect to the boundary conditions as well (loaded on the edge or in the middle). Comparative analysis was performed for full and hollow-timber cross sections made of softwood (fir). Furthermore, the analysis included hollow-timber cross-section elements made of hardwood (hornbeam). To obtain information on the base value of CSPG for full-timber cross sections made of hardwood, samples of solid hardwood were also analyzed, but only for the basic set-up, without examining the influence of the position of the load concerning the boundary conditions.

Before starting the experiment, the wood density and moisture content were measured on specially made cube specimens. A total of 24 samples were made, 12 of each type of wood (see Figure 4 and Table 1).

Regarding the geometry (see Figure 5), the cross-sectional dimensions for all specimens were 120 × 240 mm, while the length was as follows: 105, 209, 400, 440, 520, and 640 mm.

A total of 120 specimens were made, whereby 6 samples were made for each of the 20 groups. A particular group of specimens is characterized by the type of wood, the type of cross-section, and the length of the specimen, where the length of the specimen represents whether the specimen is loaded in the strong or weak axis direction. According to the schemes in Figure 6, softwood full and hollow timber cross-sections, as well as hardwood hollow timber cross-section specimens were tested (a total of 18 groups). Additionally, according to the schemes in Figure 6a,b, hardwood full-timber cross-section specimens were tested (a total of two groups).

While the hollow timber elements can be seen in the image above, the full normal ones are the same, but completely without the elliptical holes. As it can be seen in Figure 5, the first and last lamellae are serrated, so additional elements should be inserted to make the outer surfaces flat; the bottom one due to support, and the upper one due to force input. These additional elements are exactly in the same shape as lamellas P1 and P4 (Figure 5), but they are made of harder timber to avoid local embossing. Steel plates were placed on top of these additional elements, through which the load was applied (Figure 7).

### 2.3. FEM Description

The numerical analysis was performed using the Dlubal RFEM [37] software package, more precisely, the RSECTION module. A parameterized input allowed entry of the cross-section dimensions and internal forces in such a way that they depend on certain variables [38]. The objective of the numerical analysis was to make a parametric analysis, the results of which would show where the highest stresses occur. Consequently, the optimal cross-section, which is between the two extremes, the full and hollow cross-section, is determined. For the sake of simplicity and easy comparison, all variant models are loaded with a pressure of 1 N/mm^2^. The loading scheme and boundary conditions for FEM can be seen in Figure 8.

Material properties used for modeling timber elements are shown in Table 2.

## 3. Results

### 3.1. Experimental Work

Considering the large number of specimens and tests, it is not reasonable to accommodate all the graphs in this section. Therefore, only the characteristic results in form of load-displacement curves for each group of samples are presented (Figure 9), and the other values are presented in a tabular comparison. The final objective for all groups of specimens was to compare the load-carrying capacity and behavior of innovative hollow GL timber elements with normal GL timber elements. This was primarily referred to the specimens made of softwood, as a raw material more used in practice. Nevertheless, the important objective was to compare the characteristics of hollow softwood and hardwood specimens, as well as to determine the base CSPG value of the hardwood material.

Figure 9 shows the almost linear behavior of the specimens up to the yielding point, after the slope of the curve decreases, the displacement increases without an increase in force, and finally, failure of the timber occurs. Such behavior was common to all types of specimens, however, there are different failure modes for different types of specimens. In case of hollow timber specimens, failure has occurred at the weakest, or thinnest part of the cross-section, between the two elliptical cavities. In the case of normal GL timber specimens, the timber cracked when the compressive strength had been reached. 

Furthermore, softwood specimens with elliptical cavities (ME) are expected to have the lowest stiffness, which can be seen from the slope of the curve. If we compare it with normal softwood specimens, without holes (MP), stiffness and strength reached by ME specimens are significantly lower. Moreover, due to the full cross-section, the horizontal part of the curve representing MP specimens indicates greater compression ductility. The physical manifestation of this is the imprinting of load cell into the timber element. Finally, hardwood specimens reached the highest stiffness and failure force, and the cause is higher timber density, i.e., compressive strength. However, an undesirable consequence of this is a brittle fracture. Although the principles leading to the failure of hardwood specimens with elliptical cavities (TE) or without holes (TP) are like those made of softwood, another difference could be seen. The hardwood specimens cracked at the finger joint due to the high strength of the timber, greater than glue. It was especially noticed on specimens loaded at the edge and in the middle.

In addition to the comparison of the load-displacement curves, the failure modes of the specimens were analyzed and compared. The failure modes can be divided into two characteristic groups, depending on whether it represents a hollow- or full-timber cross-section. The main failure mode of the innovative hollow GL timber elements was the timber failure of the area between the holes, in the direction of the applied load Figure 10a,b). Because elliptical cavities are arranged in columns, the load transmission is simple, along the ridges of solid wood. The cracks are mainly a straight line connecting the tops of the arcs of the ellipses. This indicates a proper path of load transmission and that the failure occurred during crushing in the cavity area. In the case of normal GL timber elements, failure occurred when the compressive strength perpendicular to the grains is reached, and fracture followed the stress trajectory (Figure 10d,e). Test of compressive strength perpendicular to the grain for specimens loaded in the direction of the stronger axis (Figure 10a,b,d,e) indicated that the behavior of these specimens have been similar to specimens loaded in the direction of the weaker axis (Figure 10c,f). Again, an almost linear behavior was observed, which turned into a curve, that indicates the yielding of the material. In this case, too, it could be observed that the cracks in hollow timber specimens were predominantly vertical, in the direction of the force, connecting the cavities.

### 3.2. FEM Analysis

At the very beginning, it was important to verify the FEM analysis, in order to be able to carry out a parametric analysis for future research. For this purpose, experimentally investigated specimens were analyzed, and the result is shown in Figure 11.

As it can be seen in Table 3, f_c,90, k_ for MP-209 mm was determined to be 4.07 MPa (in the upper corner), and for ME-209 mm, f_c,90,k_ = 1.83 MPa. Those stresses initially appeared in the upper corner and on the perimeter of the holes in the case of hollow GL specimens, which was also confirmed in the FE mode (Figure 11). By evaluating the results of the FEM analysis, the initial σ_z_ stress for MP-209 mm was 4.067 MPa (Figure 11a) for the value of failure load 106.7 kN (Table 4) and 1.869 MPa (Figure 11b) for MP-209 mm specimen and the value of failure load 47.7 kN (Table 4). Furthermore, in Figure 10a,d, the failure mode and primary cracks are shown. This was also confirmed by the FEM model (check the stress trajectories in Figure 11).

In the next step, a parametric analysis was made. All results of the parametric analysis were evaluated and visualized in an appealing graphical form (Figure 12, Figure 13, Figure 14 and Figure 15). As can be seen in the figures, the analysis was carried out step by step, from the model with the highest percentage of cavities to the model without cavities.

The analysis was made with several groups of models, and all models refer to softwood. The goal is to make a comparative analysis of the stress due to the geometrical distribution of the cavities. The first group of models refers to models where cavities were gradually removed in rows, starting from the bottom lamella (Figure 12).

The next group of models (Figure 13) is reflected in the variability of the holes in alternating rows.

The third group of models (Figure 14) is shown in the variation of the columns of cavities. 

Finally, the last group of models (Figure 15) refers to specimens that are the opposite of the previous group, i.e., the holes only present in the central area, while the final model is a normal GL timber specimen, without holes.

## 4. Discussion

### 4.1. Experimental Work

The ratio of failure forces of all specimens is shown in Table 4. As expected, the groups of hardwood hollow timber specimens showed the greatest ultimate force followed by groups of softwood specimens without holes, and finally groups of softwood hollow timber specimens. The hollowed hardwood timber specimens showed a higher load capacity even than the softwood specimens without holes due to approximately three times higher CSPG. The reason that groups of softwood hollow timber specimens had the lowest load capacity lies in the small distance between the cavities, that is, the small thickness of the solid wood between the cavities, which would transmit the load from the top to the bottom of the sample.

Furthermore, compressive strengths perpendicular to the grain for each group of specimens, with associated k_c, 90_ factors are given in Table 3.

### 4.2. FEM Analysis

The first model (Figure 12a) in the first group shows the stress concentration where the maximum stress occurs in the area between the holes and is 3.118 N/mm^2^. When the first row of cavities had been removed (Figure 12b), it minimally affected the stress distribution; however, although at the bottom of the sample the stress was lower, the maximum stress was similar to the first one (3.155 N/mm^2^). By removing the holes on the next lamella (Figure 12c), the stress distribution was more favorable both locally and globally, especially at the bottom part of the specimen.

The second group of FE models shows the variability of the holes in alternating rows. Figure 13a shows the stress distribution when the cavities on each successive lamella were removed. The stress was less on the vertical timber areas between the cavities, but that is why the stress was slightly higher on the timber horizontal areas between the cavities compared to the first group of models. The second (Figure 13b) and third models in this group were very similar, although the third model (Figure 13c) in this group was slightly better due to the absence of cavities on the outer parts. The maximum stress for the first model in this group was 3.212 N/mm^2^, while the stress in the second model was 3.301 N/mm^2^.

The next group is shown in Figure 14a, where the cavities are left out in the middle cross-section area, and this was reflected in the stress distribution. The maximum stress was lower compared to the previous models (2.697 N/mm^2^) and, the stress distribution is more favorable because there are no stress concentrations in the central part. When the holes in each subsequent lamella are omitted (Figure 14b,c), the global stress distribution was more favorable, but due to the smaller number of holes, slightly higher stress occurred at the edges of the ellipse, caused by the flow of the principal stresses.

For the last group, it can be observed that the most favorable specimen in terms of stress was the just-mentioned normal specimen (Figure 15c), while the second specimen (Figure 15b) in this group showed better behavior compared to the first (Figure 15a), and the reason for this was the lower perforation of the specimen and, accordingly, less stress concentration.

## 5. Conclusions

From the presented study, it can be concluded that the CSPG of softwood, for a full laminated cross-section loaded in the direction of the stronger axis, is equal to 4.07 MPa and the CSPG of hardwood is equal to 12.96 MPa, with the coefficient k_c,90_ equal to 1.0. For load action at the edge of the element, the factor k_c,90_ = 1.24 was obtained, as lower by 20% than the value prescribed in Eurocode 5 [3] of 1.55. For the load action at the middle of the element, the factor k_c,90_ = 1.45 was obtained, which is lower by 12% than the value prescribed in [3] of 1.66. The CSPG of softwood, for a hollowed laminated cross-section loaded in the direction of the stronger axis, decreases by about 55% compared to the full cross-section, with a value of 1.83 MPa, and for hardwood, it decreases by about 50%, to a value of 6.58 MPa, with the coefficient k_c,90_ equal to 1.0. For load actions at the edge and the middle of the element, k_c,90_ factors were obtained with a value closer to those obtained for full cross-section, which indicates the same phenomenology, regardless of cross-sectional weakening.

In addition, it can be concluded that the CSPG of softwood, for a full laminated cross-section loaded in the direction of the weaker axis, is equal to 4.17 MPa and the CSPG of hardwood is equal to 15.08 MPa, with a coefficient of k_c,90_ equal to 1.0. For load action at the edge of the element, the factor k_c,90_ = 1.42 was obtained, as lower for 30% than the value prescribed in [3] of 2.07. For load action at the middle of the element, the factor k_c,90_ = 1.46 was obtained, which is lower by 35% than the value prescribed in [3] of 2.21. The CSPG of softwood, for a hollowed laminated cross-section loaded in the direction of the weaker axis, decreases by about 55% compared to the full cross-section, with a value of 1.90 MPa, and for hardwood, it decreases by about 55%, to a value of 6.75 MPa, with the coefficient k_c,90_ equal to 1.0. It can be concluded that the properties are similar to the situation when the cross-section is loaded in the direction of the stronger axis.

Moreover, it can be concluded that the degree of hollowness is proportional to the CSPG regardless of the type of wood. Moreover, the weakening does not affect the transfer of force with respect to the boundary conditions and position of the load, i.e., the k_c,90_ factors are approximately similar for hollowed and full cross-sections. However, in order to better understand it, it is necessary to study the stress distribution and force path in more detail using the DIC measurement method. As mentioned in the introduction, the factor k_c,90_ is difficult to determine unequivocally for different boundary conditions. This research presented that the values given in European standards [3] still cannot be applied uniformly. So, further research is necessary for the correction of factors regarding the type of wood, type of section, etc.

Finally, FE analysis confirmed the experimental work. The results of the comparative numerical analysis indicated how the arrangement and layout of the cavities affect the stress distribution. It has been proven that by removing certain rows or columns of holes, we can favorably influence stress distribution. If the first lamellae are full, without cavities, this has a positive effect on the overall behavior of the element, that is, it will crack at a higher force. By avoiding cavities in every subsequent lamella, stress concentration is reduced and the area between the two cavities is increased, which ultimately results in a higher load capacity of the element. If the central part of the cross-section is without holes, the stress is reduced, but special attention should be paid to the distance of the holes from the edges so that local cracking does not occur. In the end, if the cavities are present only in the central part of the element, the behavior of the element is more favorable, but the question arises as to how meaningful it is to make such patterns and how many advantages there are compared to the specimen without cavities, considering that the perforation of this kind of specimen is much lower compared to the previously studied samples. In the continuation of the research, it is planned to conduct an experimental investigation of variant solutions for innovative hollow glued laminated timber elements.

## 6. Patents

The producer of the timber elements, a company (Tersa Ltd from Croatia), is in the application process for an intellectual property patent so that this product and system are protected.

## Figures and Tables

**Figure 1 polymers-14-03403-f001:**
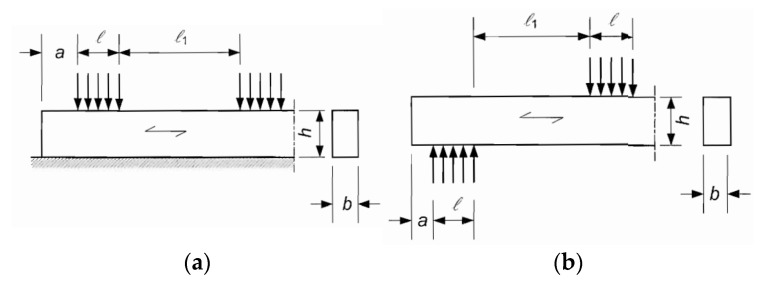
Member on: (**a**) Continuous supports; (**b**) discrete supports.

**Figure 2 polymers-14-03403-f002:**
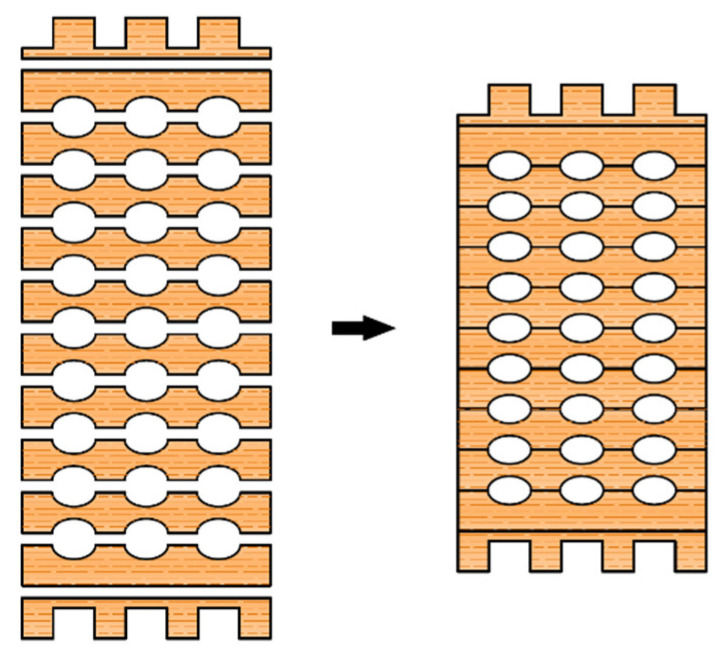
Individual lamellae of the element with elliptical holes and the cross-section of the assembled hollow glued laminated timber element.

**Figure 3 polymers-14-03403-f003:**
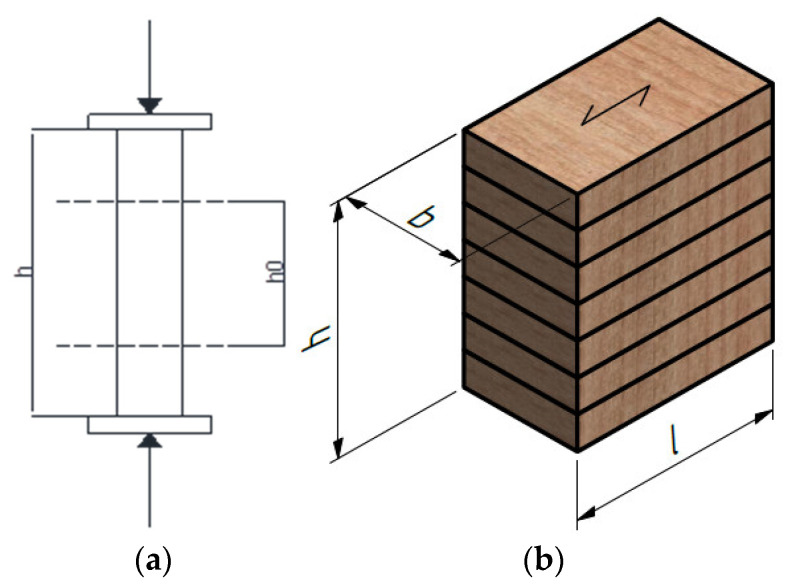
Test setup: (**a**) Load and gauge position; (**b**) specimen dimensions.

**Figure 4 polymers-14-03403-f004:**
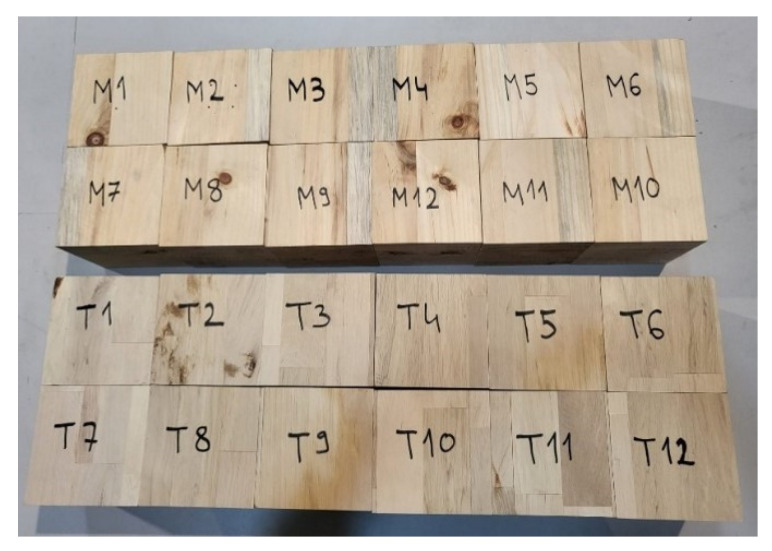
Cube specimens—density measurement.

**Figure 5 polymers-14-03403-f005:**
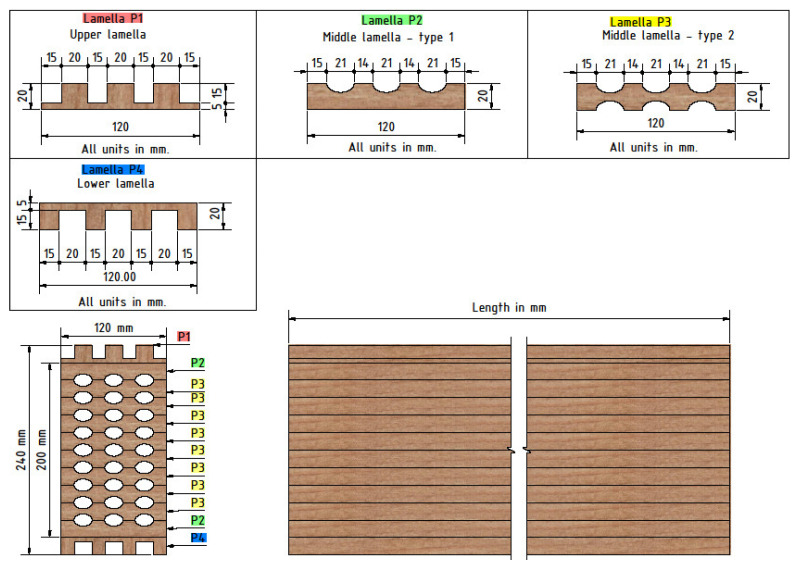
Lamellae and assembled specimen with elliptical cavities.

**Figure 6 polymers-14-03403-f006:**
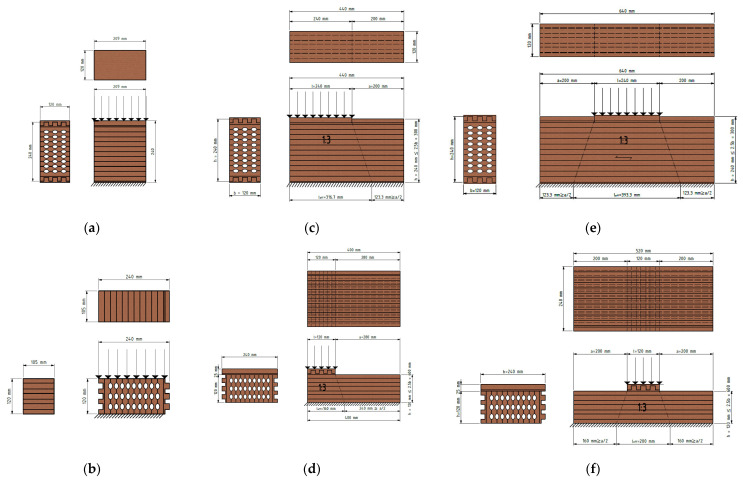
Positioning and loading of specimens, considering the axis direction and boundary conditions: (**a**) Specimen loaded over the entire surface in the strong-axis direction (four groups); (**b**) specimen loaded over the entire surface in the weak-axis direction (four groups); (**c**) specimen loaded on the edge in the strong-axis direction (three groups); (**d**) specimen loaded on the edge in the weak-axis direction (three groups); (**e**) specimen loaded in the middle area in the strong-axis direction (three groups); (**f**) specimen loaded in the middle area in the weak-axis direction (three groups).

**Figure 7 polymers-14-03403-f007:**
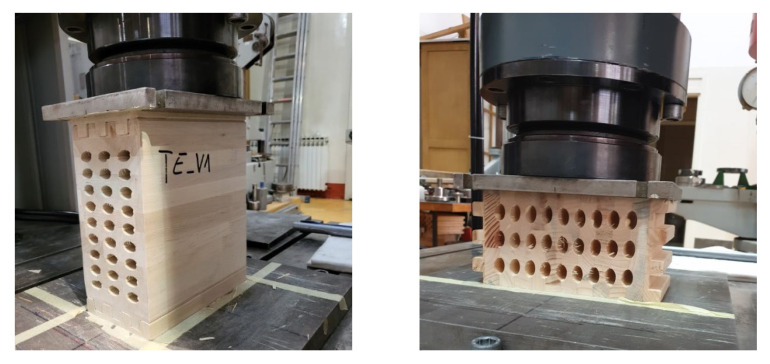
Loading of specimens: (**a**) in strong-axis direction; (**b**) in weak-axis direction.

**Figure 8 polymers-14-03403-f008:**
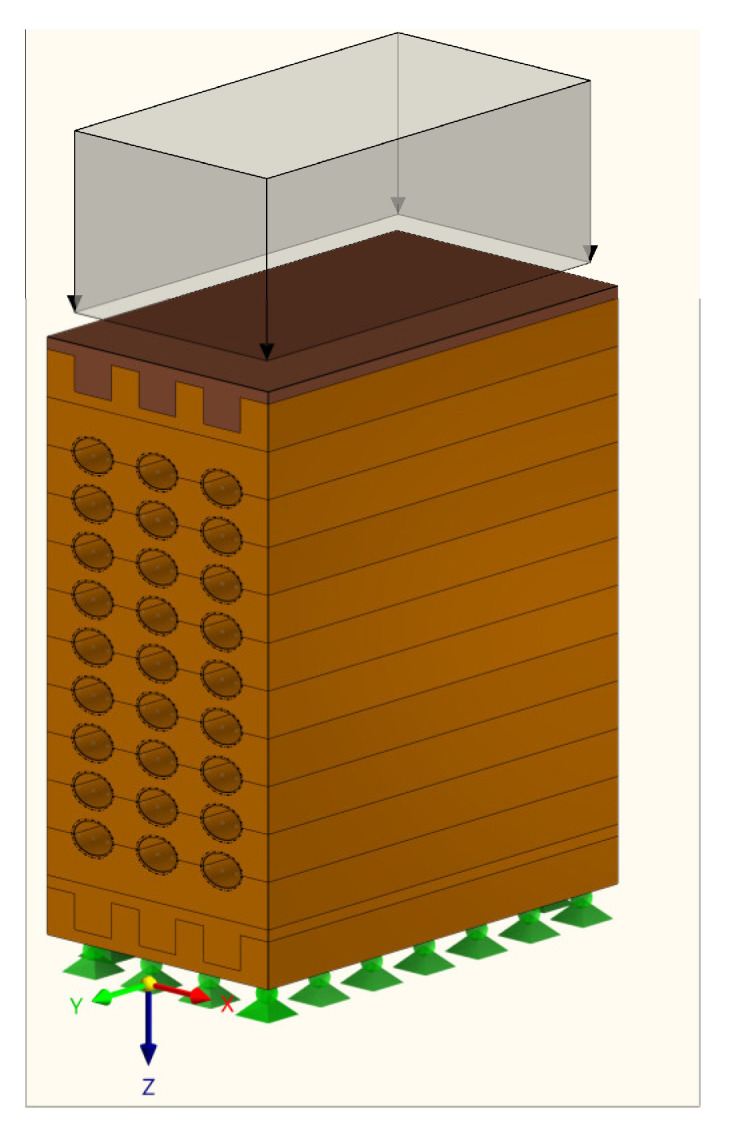
FEM—loading scheme.

**Figure 9 polymers-14-03403-f009:**
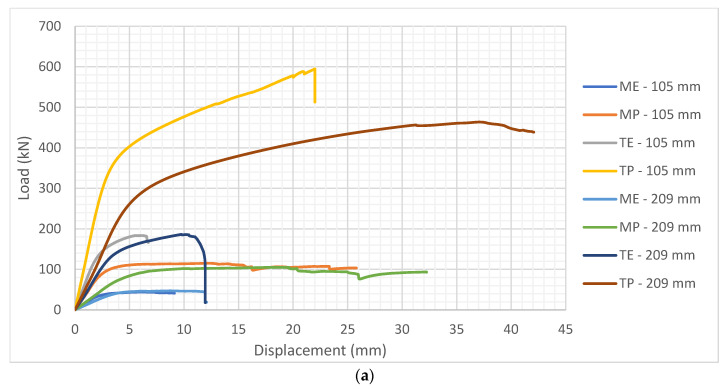
Characteristic load-displacement curves for each group of specimens: (**a**) Standardized specimens; (**b**) specimens loaded on edge of the element; (**c**) specimens loaded in the middle of the element.

**Figure 10 polymers-14-03403-f010:**
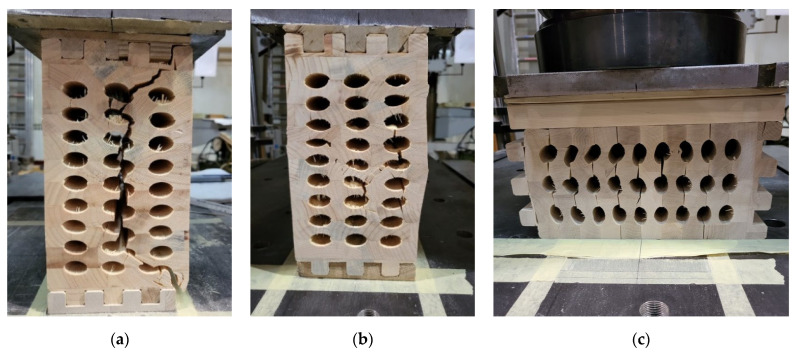
Failure modes characteristic for group of specimens: (**a**) ME-209 mm; (**b**) TE-209 mm; (**c**) ME-400 mm; (**d**) MP-209 mm; (**e**) TP-209 mm; (**f**) MP-400 mm.

**Figure 11 polymers-14-03403-f011:**
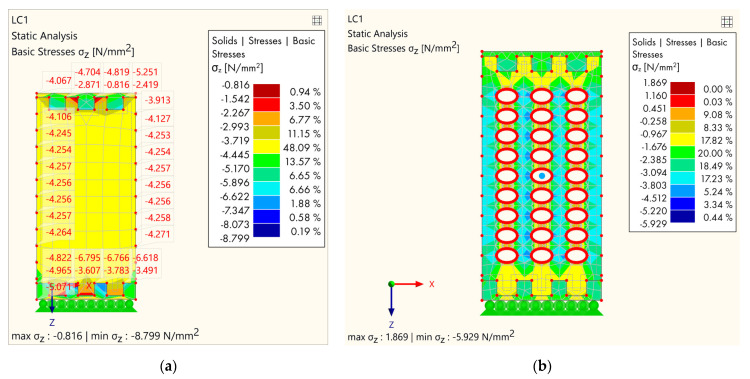
FE verification: (**a**) MP-209 mm; (**b**) ME-209 mm.

**Figure 12 polymers-14-03403-f012:**
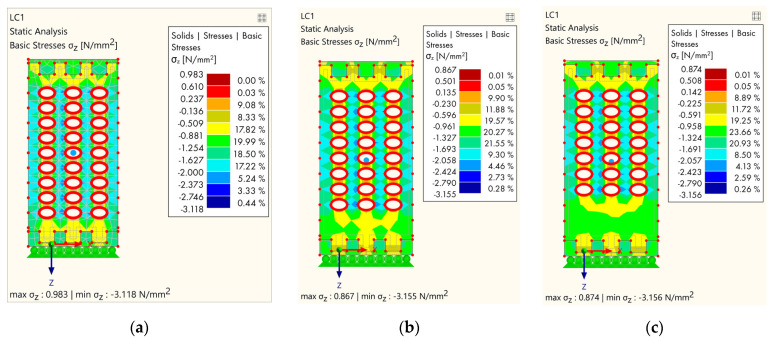
Results of FEM analysis—stress: (**a**) Completely perforated timber element; (**b**) the first lamella without cavities; (**c**) the second lamella without cavities.

**Figure 13 polymers-14-03403-f013:**
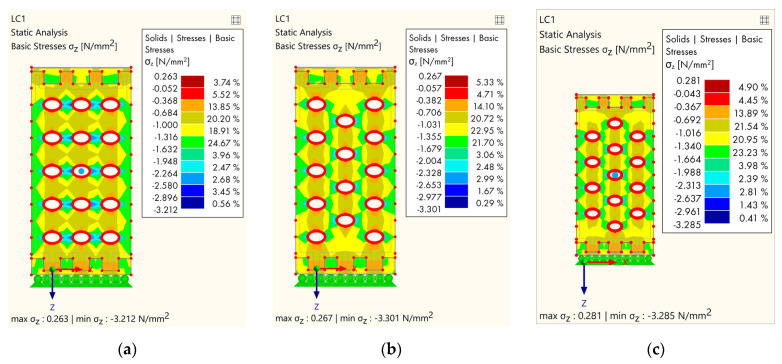
Results of FEM analysis—stress: (**a**) Each subsequent lamella without cavities; (**b**) alternating arrangement of holes, type 1; (**c**) alternating arrangement of holes, type 2.

**Figure 14 polymers-14-03403-f014:**
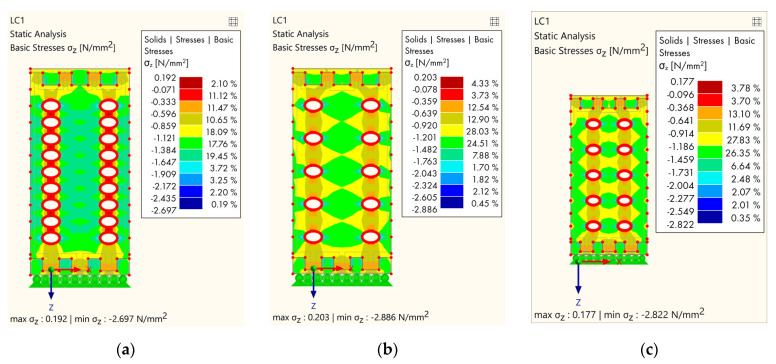
Results of FEM analysis—stress: (**a**) The central part without cavities; (**b**) the central part and each subsequent lamella without cavities; (**c**) the outer part and each subsequent lamella without holes.

**Figure 15 polymers-14-03403-f015:**
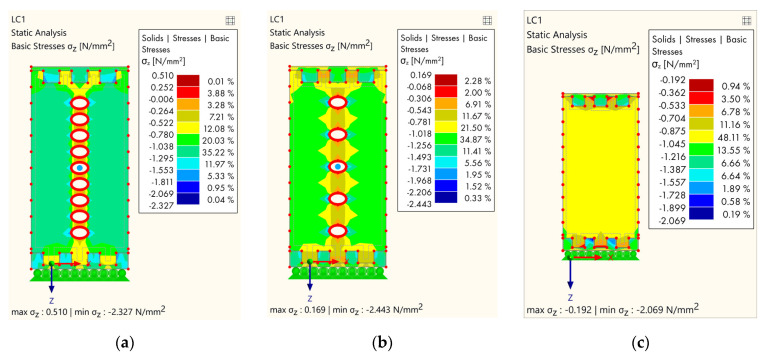
Results of FEM analysis—stress: (**a**) The central part with cavities; (**b**) the central part and each subsequent lamella with cavities; (**c**) normal—without cavities.

**Table 1 polymers-14-03403-t001:** Density of cube specimens.

	Width (mm)			Length (mm)			Height (mm)			Weight (g)			Density (kg/m^3^)		
	Avg.	CoV. (%)	St. Dev.	Avg.	CoV. (%)	St. Dev.	Avg.	CoV. (%)	St. Dev.	Avg.	CoV (%)	St. Dev.	Avg.	CoV. (%)	St. Dev.
**Softwood** M1–M12	119.3	0.19	0.22	120.1	0.30	0.36	119.0	0.22	0.27	657.9	1.52	9.99	385.7	1.42	5.47
**Hardwood** T1–T12	119.3	0.21	0.25	119.9	0.35	0.42	118.5	0.89	1.06	1337.4	1.35	18.08	789.4	1.60	12.60

**Table 2 polymers-14-03403-t002:** FEM—material properties.

Moduli	Symbol	Value (N/mm^2^)
Modulus of elasticity parallel	E_0, mean_	11,500
Modulus of elasticity perpendicular	E_90, mean_	300
Shear modulus	G_mean_	650
Modulus of elasticity parallel	E_0,05_	9600
Modulus of elasticity perpendicular	E_90,05_	250
Shear modulus	G_05_	540

**Table 3 polymers-14-03403-t003:** List of CSPGs for different load modes and boundary conditions with associated k_c,90_ factors.

Type of Cross-Section	MP		ME		TE		TP	
Specimen Length (mm)	f_c,90, k_ (MPa)	k_c, 90_	f_c,90, k_ (MPa)	k_c, 90_	f_c,90, k_ (MPa)	k_c, 90_	f_c,90, k_ (MPa)	k_c, 90_
105	4.17	1.00	1.90	1.00	6.75	1.00	15.08	1.00
209	4.07	1.00	1.83	1.00	6.58	1.00	12.96	1.00
400	5.90	1.42	2.67	1.40	9.59	1.42	/	/
440	5.03	1.24	5.90	1.45	8.16	1.24	/	/
520	6.08	1.46	2.95	1.55	10.42	1.54	/	/
640	5.90	1.45	2.78	1.51	9.72	1.48	/	/

ME—softwood hollow, MP—softwood normal, TE—hardwood hollow, TP—hardwood normal.

**Table 4 polymers-14-03403-t004:** Failure force—comparison.

Specimen Length (mm)	Type of Cross-Section	Average Failure Force (kN)	CoV. (%)	St. Dev.	F_max_-Ratio in Relation to ME	F_max_-Ratio in Relation to TE
105	ME	45.2	9.07	4.1	1.00	0.54
	MP	118.4	7.09	8.4	2.62	1.41
	TE	185.2	8.26	15.3	4.10	2.21
	TP	622.19	6.22	38.7	13.77	7.43
209	ME	47.7	5.03	2.4	1.00	0.54
	MP	106.7	9.09	9.7	2.24	1.22
	TE	184.1	1.90	3.5	3.86	2.10
	TP	453.80	2.29	10.4	9.51	5.17
400	ME	87.3	9.51	8.3	1.00	0.31
	MP	219.3	8.76	19.2	2.51	0.78
	TE	320.2	6.09	19.5	3.67	1.15
	TP	/	/	/	/	/
440	ME	75.3	1.73	1.3	1.00	0.39
	MP	157.4	8.64	13.6	2.09	0.82
	TE	255.3	10.77	27.5	3.39	1.33
	TP	/	/	/	/	/
520	ME	126.5	5.06	6.4	1.00	0.24
	MP	282.4	5.24	14.8	2.23	0.54
	TE	413.7	3.50	14.5	3.27	0.79
	TP	/	/	/	/	/
640	ME	106.9	8.70	9.3	1.00	0.31
	MP	196.0	4.23	8.3	1.83	0.57
	TE	323.8	6.61	21.4	3.03	0.94
	TP	/	/	/	/	/

ME—softwood hollow, MP—softwood normal, TE—hardwood hollow, TP—hardwood normal.

## Data Availability

Data available on request due to restrictions, e.g., privacy or ethics. The data presented in this study are available on request from the corresponding author.

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
