# Peer review of "Compressive Strength Properties Perpendicular to the Grain of Hollow Glue-Laminated Timber Elements"

_polymers, 2022, doi:10.3390/polym14163403_

Round 1

Reviewer 1 Report

P1 L11: Present and past tenses are mixed. Recommend to maintain a fixed tense along the manuscript.

P1 L22: Present and past tenses are mixed. Recommend to maintain a fixed tense along the manuscript.

< Materials and Methods>

P5 L200 Table 1 should be explained more clearly.

<Results>

P10 Figure 8 The background grid lines are cluttered.

P11 The full cross-section of hardwood specimen - Load-displacement curve should be added.

P12 Figure 11 The stiffness of full cross-section of hardwood should be added.

P14 Figure 13 The stress concentration point calculated by finite element method is not consistent with the actual experimental cracks.

Author Response

Dear reviewer,

Thank you very much for your valuable comments, suggestions and help.

Please find the answers attached.

Kind regards,

Author

Reviewer 2 Report

The authors studied compressive strength perpendicular to the grain of a new wooded product – hollow glulam, determined the characteristic value of the compression strength, and evaluated a load configuration factor. Moreover, the authors run optimization FEM analysis using different cross-sectional hollow configurations that help to perfect the hollow cross-sectional setup. I recommend the paper for publication. It brings new knowledge to the product that has potential in structural applications.

Few comments:

The paper holds few descriptions of EN 408 (Table 1, Figure 1, Figure 2, …) It is common knowledge, and it is enough to cite EN 408. It will save space that has to be used in other sections.

Abstract

Please shorten the part “why and how” and expand the part “main outputs”. The abstract should have the main results including results of FEM optimization of the product.

Introduction

The main aim (row 60) and objectives (row 140…) are mentioned in two separate places. Please, join this part together.

Objectives in rows 149-174 could be shortened. There is no need to describe the method or materials in the objectives. If the DIC was not part of the study, why mention DIC in row 173? Authors justified metal-on-wood compression setup in rows 168-170. The authors used other wooded elements in the test setup (raw 168-170) thus they used wood-to-wood contact. Please, describe this contradiction or remove the statement.

Materials and Methods

The method is not clearly described. How many samples were loaded on the edge, how many were loaded in the middle, and how many were loaded vertically and horizontally? Is vertical/horizontal load related to the strong and weak axis? A simple table describing the experimental setup will be useful. It is too late to get partial information from figures in Results or descriptions in Conclusions.

Table 2 belongs to the results.

Why the longer hornbeam samples, above 400 mm long were not evaluated, according to Table 4?

Please define contact loading areas for vertically and horizontally loaded samples.

Please move the FEM method description to Materials and Methods.

Please, list the wood material properties used in the FEM model. Also, please supply a loading scheme of the FEM model.

The FEM analysis description in rows 337 – 347 is overused. Please, focus on the tools and setting used in this study. There is no need to describe the functionality of the Dlubal RFEM software.

Keep the results of FEM in sections Results. It is not clear what kind of stress is shown in Figures 13, 14, 15, and 16. According to the aim of this study (row 61) authors focus on compression stress. Stress txz seems to be shear stress. The force Vz is perpendicular to the grain and loaded surface x is the anatomical cross-section. Figure 16 c proves that the stress txz is not the compression stress. According to the coordinate system of the FEM model, authors could show results of szz or principal stress or von Mises stress. Otherwise, please, give a reason for txz.

Discussion about FEM results should be in a part Discussion.

Each FEM model requires verification. If the DIC method is not available, the force-displacement data below the proportional limit could be used for verification.

Results

One force-displacement figure is enough (Figures 8, 9, and 10 are redundant). There is no need to show more examples. A table with results of force-displacement slopes including standard deviations gives better insight and this information is missing.

Results do not fully reflect the method. TP data are missing, for example, a TP curve should be also in Figure 11. Table 3 misses data on TP average failure force. Please, supply also the standard deviation of average failure forces.

Fmax percentage of TE cannot be 100% in each group. The failure force of TE should be different from the failure force of TP!

The summary results (table 4) are vague. Looks like the results are the characteristic value of all grouped samples. Results of samples loaded horizontally, vertically, loaded in the middle, or on the edges are missing. Maybe they are not. Please be more specific. A reader does not understand the full story when the summary table is given only, or the Method does not describe an experimental setup properly, and a reader does not know what to expect. Please, supply the missing links or explain.

Raw 440: Do the authors mean the “weaker” axis?

This weaker and strong axis terminology is messy. Figure 12 (c and f) refers to a stronger axis for the length of 400 mm (rows 324-325) when a sample is flat loaded. Row 204 describes a strong axis in the direction of h in Figure 4. Please be consistent. In the materials and method section, there must be a clear explanation.

A formal comment

Please define the abbreviation “TP”.

Overall review

This study gives insight into the novelty material of engineering wood products. It supplies good knowledge, but the paper description could be shortened to the common knowledge and expanded in detail in method, results, and discussion. So, the paper needs to be improved.

Author Response

Dear reviewer,

Thank you very much for your valuable comments, suggestions, and help.

It improved our manuscript!

Please find the answers attached.

Kind regards,

Author

Round 2

Reviewer 1 Report

This paper can be accepted.